# Potential Dietary and Therapeutic Strategies Involving Indole-3-Carbinole in Preclinical Models of Intestinal Inflammation

**DOI:** 10.3390/nu15234980

**Published:** 2023-11-30

**Authors:** Aisha Qazi, Shane Comiskey, Nathan Calzadilla, Fatimah Amin, Anchal Sharma, Ei Khin, Nathaniel Holton, Christopher R. Weber, Seema Saksena, Anoop Kumar, Waddah A. Alrefai, Ravinder K. Gill

**Affiliations:** 1Division of Gastroenterology and Hepatology, University of Illinois at Chicago, Chicago, IL 60612, USA; aqazi7@uic.edu (A.Q.); shanec@uic.edu (S.C.); famin3@uic.edu (F.A.); anchals@uic.edu (A.S.); eik2@uic.edu (E.K.); nwholton82@gmail.com (N.H.); saksena@uic.edu (S.S.); anoop@uic.edu (A.K.); walrefai@uic.edu (W.A.A.); 2Department of Bioengineering, University of Illinois at Chicago, Chicago, IL 60612, USA; ncalza3@uic.edu; 3Department of Pathology, University of Chicago, Chicago, IL 60637, USA; cweber@bsd.uchicago.edu; 4Jesse Brown VA Medical Center, Chicago, IL 60612, USA

**Keywords:** AhR, chronic DSS, ileitis, colitis, IEC, gut microbiota

## Abstract

Diet–microbiota interactions are emerging as important contributors in the pathogenesis of inflammatory bowel diseases (IBD), characterized by chronic inflammation of the GI tract. The aryl hydrocarbon receptor (AhR) transcription factor regulates xenobiotic metabolism and is activated by exogenous ligands, including indole-3-carbinole (I3C), which is found in cruciferous vegetables. However, studies investigating the impact of dietary I3C and AhR in preclinical models resembling human IBD are lacking. Mice (WT or AhR KO in IECs, 6–8 weeks) or SAMP/YitFC and AKR/J control (4 weeks, m/f) were fed an AhR ligand-depleted or I3C (200 ppm)-supplemented diet. There were increased levels of LPS and exacerbated inflammation, resulting in increased mortality in AhR^ΔIEC^ mice fed the AhR ligand-depleted diet in response to chronic DSS. The mechanisms underlying the protective effects of I3C supplementation during colonic colitis involved amelioration of intestinal inflammation and restoration of the altered gut microbiota, particularly the families of clostridicae and lachnospriaceae. Furthermore, the AhR-depleted diet led to the emergence of pathobiont *Parvibacter caecicola* in WT mice. SAMP/YitFc mice with spontaneous ileitis showed significant recovery in epithelial abnormalities when fed dietary I3C. These data demonstrate the critical role of AhR and the mechanisms of dietary I3C in maintaining epithelial homeostasis and ameliorating inflammation.

## 1. Introduction

Inflammatory bowel diseases (IBDs), including Crohn’s Disease (CD) and ulcerative colitis (UC), are characterized by chronic inflammation of the gastrointestinal tract [1]. Though the exact cause of IBD is unknown, its pathophysiology is multifactorial, with genetic predispositions, immune dysregulation, gut dysbiosis, and environmental factors such as diet playing a role in the onset and progression of these diseases [1]. Current therapeutic strategies for the treatment of IBD symptoms include anti-inflammatory aminosalicylates, immunosuppressives, antibiotics, TNF-antagonists, the recently approved integrin alpha4beta7 antagonist, and Janus Kinase (JAK) inhibitor [2]. However, each of these drugs yield moderate success rates with significant side effects [3]. Recently, the aryl hydrocarbon receptor (AhR) has gained recognition as a potential therapeutic target for IBD after being identified as an IBD susceptibility gene in a genome-wide association study (GWAS) [4]. Activation of AhR is crucial for xenobiotic detoxification and the maintenance of intestinal homeostasis [5]. After ligand binding, AhR, which is present in the cytoplasm, codimerizes with the AhR nuclear translocator (ARNT) and moves to the nucleus to bind to the xenobiotic response element (XRE) on DNA, resulting in the induction of xenobiotic metabolizing enzymes, Cytochrome P450 Family 1 Subfamily A Member 1 (CYP1A1) and CYP1B1, as well as Interleukin-10 (IL-10) and IL-22 [6]. As AhR is ubiquitously expressed, previous studies have interrogated the cell-specific roles of AhR in modulating acute intestinal inflammation [7,8,9]. These studies demonstrated worsened inflammation in mice with intestinal epithelial cell (IEC)-specific constitutive deletion of AhR in acute colitis [8,9]. Consistently, activation of AhR has been shown to be beneficial in models of acute inflammation [10,11,12,13]. However, the therapeutic potential of AhR and its dietary ligands in preclinical models resembling the chronic nature of human IBD has not been systematically investigated. Glucobrasscin found in Brassicaceae family vegetables such as broccoli, cabbage, and brussels sprouts is broken down into indole-3-carbinole (I3C), a well-recognized dietary component and AhR agonist [14]. Notably, the gastric acid breakdown of I3C also results in the products 3,3′-diindolylmethane (DIM) and indole [3,2-b] carbazole (ICZ), potent agonists of AhR [14]. A recent study showed intraperitoneal administration of I3C reduced intestinal inflammation in a mouse model of TNBS-induced colitis via alterations in the gut microbiota [15]. Similarly, other dietary AhR ligands such as tryptophan and β-napthoflavone (BNF) have been shown to attenuate intestinal inflammation [16,17]. However, an in-depth analysis of how I3C through dietary consumption impacts ulcerative colitis as well as ileitis resembling CD has not yet been investigated. Given the translational potential of dietary modifications for IBD, we aimed to investigate the impact of inducible deletion of AhR from IECs and dietary I3C manipulations in models of IBD resembling human IBD. Our data strongly highlight how diet can shape the inflammatory response and microbiota, as well as emphasizing how nutritional strategies can be exploited to improve chronic intestinal inflammation.

## 2. Materials and Methods

### 2.1. In Vivo Studies

Prior approval was obtained for performing in vivo studies in C57BL/6, AhR^fl/fl^, AhR^ΔIEC^, AKR/J, and SAMP1/YitFc mice from the Institutional Animal Care Committee of Jesse Brown Veterans Affairs Medical Center and the University of Illinois at Chicago. AhR^fl/fl^ mice and Villin-Cre^ERT2^ (C57BL/6 background) were obtained from Jackson Laboratory (Bar Harbor, ME) and were crossbred to generate AhR^fl/fl^ x Villin-Cre^ERT2^ mice for the tamoxifen-inducible knockdown of AhR in IECs only. Mice were weaned at 28 days and tail snipped for genotyping for homozygous expression of AhR^fl/fl^ transgene and the heterozygous Villin-Cre^ERT2^ transgene, as directed by Jackson Laboratory. Deletion of IEC AhR was induced by intraperitoneal (IP) injections of tamoxifen in corn oil (2 mg) (Sigma-Aldrich, St. Louis, MO, USA) for five consecutive days. Mice were re-injected with tamoxifen or corn oil for five days, every 2 weeks, as previously described [18]. Mice were euthanized within 2 weeks of last injections. 

AKR/J and SAMP1/YitFc mice were obtained from Jackson Laboratory at 8 weeks of age and were kept until 20 weeks of age. Mice were euthanized by CO_2_ inhalation and subsequent cervical dislocation. Upon euthanasia, blood, intestinal tissue, intestinal mucosal scrapings, feces, cecal contents, and spleen were collected. Colons were measured (cm) and weighed (g). 

### 2.2. Dextran Sulfate Sodium (DSS) Treatment 

To induce chronic colitis, C57BL/6, AhR^fl/fl^, and AhR^ΔIEC^ mice received 1.0–2.5% DSS (wt/vol) (MP Biomedical, Solon, OH, USA) in drinking water for five weeks, alternating weekly with tap water, or just tap water for all five weeks. DSS was changed every 3–4 days. Weight was monitored every other day, and mice were euthanized if they lost > 20% body weight from the start of the experiment. At the end of the 5 weeks, mice were euthanized. 

### 2.3. Diet Studies

C57BL/6, AhR^fl/fl^, AhR^ΔIEC^, AKR/J, and SAMP1/YitFc mice were fed either AIN-76A (Research Diets Inc., New Brunswick, NJ, USA, diet purified of all AhR ligands) or AIN-76A supplemented with 200 ppm indole-3-carbinole (Research Diets Inc., New Brunswick, NJ, USA). C57BL/6, AhR^fl/fl^, and AhR^ΔIEC^ were acclimated to the diets one week before the start of tamoxifen injections or DSS treatments. AKR/J and SAMP1/YitFc mice began diet at 8 weeks of age and continued for 12 weeks until 20 weeks of age. 

### 2.4. In Vitro Experiments

Caco-2 cells (ATCC, Manassas, VA, USA) were grown routinely in Eagles minimum essential media (EMEM) supplemented with 10% FBS. All cell culture medium used for maintenance contained 100 IU/mL penicillin and 100 μg/mL streptomycin. Caco-2 cells were maintained in 5% CO_2_ at 37 °C in 75 cm^2^ or 150 cm^2^ flasks and were split using 0.25% trypsin-EDTA. In preparation for treatment, Caco-2 cells were plated on 0.4 μm microporous transwells (Corning, Corning, NY, USA) in 12-well plates. Caco-2 cells were maintained for >14 days post-plating to allow for differentiation before treatments were performed. Cells between passages 25 and 45 were used for experiments. 

### 2.5. Cell Treatments

Caco-2 cells plated in 0.4μm microporous transwells (Corning, Corning, NY, USA) were serum starved for 2 h prior to treatment in 0.1% BSA. Cells were treated in the basolateral compartment with a cytokine cocktail of IL-1β (50 ng/mL), IFNγ (30 ng/μL), and TNFα (50 ng/mL) or the vehicle control. In the apical compartment, cells were either treated with indole [3,2-b] carbazole (ICZ, 50nM) (Santa Cruz Biotechnology, Dallas, TX, USA) or the vehicle control. Cell treatment was 24 h at 37 °C, 5% CO_2_. After treatment, transepithelial electrical resistance (TEER) was evaluated using EVOM3 (World Precision Instruments, Sarasota, FL, USA). Cells were then washed with 1X PBS, and filters were stored in RLT and 2-mercaptoethanol (BME) at −80 °C until RNA extractions. 

### 2.6. Histology

Distal colon or ileal tissue was fixed in 10% Formalin, embedded in paraffin, cut in 5-micron sections, stained with hematoxylin and eosin (H&E), and imaged using a Zeiss microscope (Jena, Germany). Stained slides were blindly scored by an intestinal pathologist (University of Chicago). 

### 2.7. Myeloperoxidase (MPO) Assay

Distal colon tissue was collected and homogenized in hexadecyltrimethylammonium bromide (HTAB) based on tissue weight. Neutrophil infiltration was determined by reading the absorbance at 450 nm every 5 min after H_2_O_2_ and o-dianisidine were added to lysates. 

### 2.8. Measuring Intestinal Permeability

FITC-dextran (4 kDa) (Sigma-Aldrich, St. Louis, MO, USA) was gavaged (600 mg/kg) 4 h before sacrifice. Blood was collected after euthanasia by cardiac puncture and placed in light-protected blood tubes. Blood was centrifuged at 4000–5000 rpm at 4 °C for 10 min, and the serum (supernatant) was collected. Serum FITC-dextran was measured at 490 nm and 530 nm. 

### 2.9. Immunofluorescence Staining

Ileum and distal colon tissues were embedded in optimal cutting temperature (OCT) compound and frozen in liquid nitrogen. OCT blocks were cut in 5-micron sections and stained with anti-Lyz1 and Anti-Occludin (Invitrogen, Carlsbad, CA, USA) antibodies. Slides were mounted with Dapi to stain for nuclei and imaged using a Zeiss microscope. 

### 2.10. RNA Extraction and Gene Expression

Trizol was used to extract RNA from mouse distal colon, ileum, and spleen mucosal scrapings. Chloroform was then added for phase separation, and the aqueous phase was collected and cleaned using RNA spin columns. Equal amounts of RNA from each sample were reverse-transcribed and amplified using Brilliant III Ultra-fast SYBR green RT-PCR master mix kit (Agilent, Santa Clara, CA, USA) using AriaMx 3000 (Agilent). The gene-specific primers for mouse AhR, CYP1A1, CYP1B1, AhRR, IL-10, IL-22, TNFα, IL-1β, IFNγ, CxCL2, and CCL20 were acquired through the Nucleotide Basic Local Alignment Search Tool (BLAST) (NCBI, NIH); these are mentioned in below in Table 1. 

RLT buffer (Qiagen) and 2-mercaptoethanol (BME) were used for RNA extractions from Caco-2 cells plated on microporous filters. Cells were squeezed from filters and homogenized using 1 mL syringes. Homogenates were cleaned with RW1 and RPE buffers (Qiagen) in RNA spin columns and resuspended in water for use. 

### 2.11. Western Blotting

RIPA lysis buffer (Cell Signaling, Danvers, MA, USA) was utilized for obtaining protein lysates from intestinal mucosa. Protein lysates were separated on 4–15% or 10% SDS-PAGE gels, transferred onto nitrocellulose membranes, and probed with anti-TJ/AJ, ERK, and STAT3 antibodies, followed by visualization with enhanced chemiluminescence detection reagent [19]. Quantification of band intensity was determined using ImageJ software (Version 1.53) (NIH, Bethesda, MD, USA). Anti-Occludin, Claudin-2, ZO-1, E-Cadherin, *p*-ERK, Total ERK, *p*-STAT3, and Total STAT3 antibodies were obtained from Invitrogen (Carlsbad, CA, USA) and Cell Signaling (Danvers, MA, USA) [20]. 

### 2.12. Microbiota and Bioinformatics

Cecal contents were collected, snap frozen in liquid nitrogen, and sent to Zymo Research (Irvine, CA, USA) for DNA purification and 16S sequencing. The total abundance of bacteria at each taxon was quantified. α diversity, β diversity, and principal component analysis (PCoA) were used for analysis. Amplicon sequence variant (ASV) analysis and linear discriminant analysis (LDA) were performed by DADA2 and bioBakery, respectively. 

### 2.13. Statistical Analysis 

All data presented are mean ± SEM. One-way ANOVA, student’s *t*-test, or Pearson’s correlation was utilized for statistical analysis. *p* ≤ 0.05 and *r* ≥ 0.7 was considered statistically significant. Analyses were performed using GraphPad Prism (Prism 9). 

## 3. Results

### 3.1. Deletion of AhR Specifically in IECs in Mice Induced with Chronic Colitis Results in Worsened Inflammatory Phenotype 

As the intestine has a diverse cell population, and AhR is ubiquitous, we first aimed to investigate the role of AhR in IECs in chronic intestinal inflammation using the inducible IEC-specific AhR KO mouse model (AhR^ΔIEC^). AhR deletion was confirmed by RT-PCR from intestinal mucosa (Figure 1A) and isolated IECs (Appendix A). AhR mRNA levels were significantly decreased in the distal colon mucosa of AhR^ΔIEC^ mice and markedly knocked down in IECs (Appendix A) compared to WT mice. AhR functions as a transcription factor to regulate the expression of various canonical targets. DSS treatment in WT mice increased mRNA levels of the canonical AhR target genes, CYP1A1 and CYP1B1 (Appendix A). However, this DSS-induced increase was blunted in AhR^ΔIEC^ mice treated with DSS. The AhR repressor (AhRR), present in immune cells, functions in a negative feedback fashion to regulate AhR activation and was not altered in AhR^ΔIEC^ mice with or without inflammation (Appendix A). Furthermore, anti-inflammatory cytokines and AhR target genes IL-10 and IL-22 were also suppressed in AhR^ΔIEC^ mice treated with chronic DSS compared with WT mice treated with DSS (Appendix A). Together, these data confirm the inducible knockdown of AhR in IECs.

Next, the severity of DSS-induced chronic inflammation in the colons of WT and AhR^ΔIEC^ mice was examined. Notably, we observed drastically shorter colons and larger spleens in AhR^ΔIEC^ mice treated with DSS than in the WT mice treated with DSS (Figure 1B–D). Strikingly, the spleen/body weight ratio significantly increased in AhR^ΔIEC^ mice treated with DSS compared to treated WT mice (Figure 1D). Both the WT and AhR^ΔIEC^ mice treated with DSS had similar increases in the colon weight/length ratio indicative of diarrhea (Figure 1E). All together, these phenotypes indicate worsened response to DSS-induced chronic colitis in AhR^ΔIEC^ mice compared to WT. 

### 3.2. Evaluation of Histological Scoring and Inflammatory Markers in AhR^ΔIEC^ Mice with Chronic DSS Treatment

We next examined the histological features of the distal colon via H&E staining. Histology displayed complete destruction of villus architecture and neutrophil infiltration in AhR^ΔIEC^ mice treated with DSS (Figure 2A). Blinded scoring revealed that AhR^ΔIEC^ mice with chronic colitis had significantly higher histological scores than WT mice with chronic colitis (Figure 2B). Similarly, myeloperoxidase (MPO) activity in the distal colon of AhR^ΔIEC^ mice treated with DSS was significantly higher than WT mice treated with DSS (Figure 2C). Further, the mRNA expression of proinflammatory cytokines IL-1β (Figure 2D) and IFNγ (Figure 2E) in distal colon mucosa was markedly upregulated in AhR^ΔIEC^ mice treated with DSS, which was not seen in WT mice with DSS treatment. However, mRNA expression of chemokine CXCL2 was not altered between the groups (Appendix A). Collectively, these data indicate that the removal of AhR from IECs exacerbates DSS-induced chronic colitis. 

### 3.3. Phosphorylation of ERK Is Upregulated in AhR^ΔIEC^ Mice Treated with Chronic DSS Treatment

To gain insight into the mechanisms underlying the increased inflammation seen in AhR^ΔIEC^ mice treated with chronic DSS, we examined upstream factors associated with inflammation. STAT3 is activated by phosphorylation and is necessary for proinflammatory cytokine signaling, including IL-6 and T_h_17 cell function [21]. We examined total STAT3 and phosphorylated STAT3 (*p*-STAT3) protein levels in the colon. As expected, DSS treatment significantly increased the levels of *p*-STAT3 normalized to total STAT3 in both WT and AhR^ΔIEC^ mice (Figure 3A). However, this induction in *p*-STAT3 levels was not markedly different in AhR^ΔIEC^ mice compared to WT mice (Figure 3A). The upstream ERK family of MAP kinases were next examined, as increases in phosphorylated ERK (*p*-ERK) have been established in patients with UC [22]. A significant increase in *p*-ERK normalized to total ERK protein levels was seen in AhR^ΔIEC^ mice treated with DSS compared with WT mice treated with DSS (Figure 3B). These data demonstrate that inducible deletion of AhR from IECs exacerbated intestinal inflammation in chronic DSS-induced colitis via a STAT3-independent mechanism and is associated with increased levels of phosphorylated ERK. 

### 3.4. Intestinal Permeability in AhR^ΔIEC^ Mice with and without Chronic Colitis

Because impaired intestinal barrier function is a hallmark of colitis, we next examined intestinal permeability in WT and AhR^ΔIEC^ mice induced with chronic colitis by FITC-dextran flux. No differences were found in serum levels of FITC-dextran between WT and AhR^ΔIEC^ mice treated with chronic DSS (Figure 4A). Interestingly, the levels of lipopolysaccharides (LPS), a gram-negative bacterium, was significantly higher in the serum in AhR^ΔIEC^ mice chronically treated with chronic DSS than in WT mice treated with DSS (Figure 4B), alluding to a strong role of the gut microbiota. We further examined the impact of the deletion of AhR in IECs on tight junction integrity. As expected, a significant decrease in Occludin protein levels in WT mice after chronic DSS treatment was observed (Figure 4C); however, this decrease was not significantly different than DSS-treated AhR^ΔIEC^ mice. Similarly, Occludin expression or localization by immunostaining was not further impacted by the deletion of AhR IECs with or without DSS (Appendix A). Protein expression of Claudin-2 had similar significant increases in WT and AhR^ΔIEC^ mice treated with DSS compared with mice given water (Figure 4D). These data illustrate that inducible deletion of AhR from IECs does not further alter intestinal permeability or tight junction integrity in chronic DSS-colitis; however, it causes an increase in serum LPS. This increase in serum LPS suggests that AhR^ΔIEC^ mice present with a combination of a damaged epithelial barrier and altered microbial species or metabolites. Given the importance of AhR in impacting chronic colitis, we next asked whether the increased severity of chronic inflammation in AhR^ΔIEC^ mice can be rescued by dietary supplementation of AhR ligands. 

### 3.5. Depletion of Dietary I3C Is Fatal in AhR^ΔIEC^ Mice and Worsens Chronic Colitis in C57BL/6 Mice 

Indole-3-carbinole (I3C) is derived from cruciferous vegetables such as broccoli and brussels sprouts. I3C can be further broken down by gastric acid into indole [3,2-b] carbazole (ICZ) and 3′3-diindolylmethane (DIM) [14], both potent AhR ligands, making I3C the ideal supplement to investigate AhR activation after deletion of IEC AhR. WT and AhR^ΔIEC^ mice were either fed a purified diet depleted of all AhR ligands (AIN-76A, hereby called −I3C) or a purified diet supplemented with 200 ppm I3C (+I3C). Interestingly, we found that WT mice induced with chronic colitis and fed +I3C had 100 percent survival rate (Figure 5A; green), and survival decreased to approximately 90 percent for mice depleted of dietary AhR ligands (Figure 5A; purple). However, AhR^ΔIEC^ mice fed the −I3C diet had a 0 percent chance of survival by the fourth day (Figure 5A; red). Interestingly, AhR^ΔIEC^ mice supplemented with I3C had an increased survival rate of 30 percent (Figure 5A; blue). These data demonstrated the beneficial and protective effects of dietary AhR ligands and AhR activation in IECs in chronic intestinal inflammation. Given the high mortality of AhR^ΔIEC^ mice with depletion of I3C and treated with DSS, we chose wild-type (WT) mice to further understand the mechanisms of how dietary I3C decreases the severity of chronic colitis. 

Upon inspection of the colon in WT mice with chronic DSS treatment (2.5%), we observed shorter colons and more diarrhea in mice fed the −I3C (depleted) diet, which appeared attenuated in mice fed the +I3C (supplemented) diet (Figure 5B). Quantification confirmed that mice fed −I3C and induced with colitis had significantly shorter colon lengths when compared to their control (Figure 5C), whereas colons from mice supplemented with dietary I3C and induced with chronic colitis were not significantly decreased from their control. Moreover, mice induced with chronic colitis and fed −I3C diet had significantly more diarrhea (colon weight/length) compared with mice fed +I3C diet (Figure 5D). Interestingly, the spleen size was also significantly enlarged in mice induced with chronic colitis and depleted of dietary I3C, and this enlargement was attenuated with I3C supplementation (Figure 5E). Histology revealed that the supplementation of I3C alleviated the necrosis, neutrophil infiltration, disruption of architecture, and erosion seen in DSS mice fed −I3C diet (Figure 5F). This is supported by blinded scores that show significantly higher histological scores in mice fed −I3C and treated with chronic DSS compared with their controls (Figure 5G). These data indicate that dietary AhR ligands attenuate histological features of chronic intestinal inflammation.

### 3.6. Supplementation of I3C Attenuates Chronic Colitis in WT Mice 

To understand the mechanisms of dietary I3C in protection against colitis, we first evaluated AhR activation by +I3C diet by assessing mRNA levels of the AhR target gene, CYP1A1. Data presented in Figure 6A showed a significant increase in CYP1A1 mRNA levels in mice fed +I3C compared to those fed with a diet depleted of I3C. On the other hand, the mRNA expression of AhRR remained unaltered (Figure 6B). We further characterized the level of inflammation by mRNA levels of IL-1β and TNFα. The data demonstrated a significant increase in IL-1β expression (Figure 6C) in mice induced with chronic colitis and fed −I3C, and this increase was attenuated with the addition of dietary I3C. Similarly, TNFα (Figure 6D) mRNA levels were significantly decreased in mice fed +I3C and treated with chronic DSS compared with mice fed −I3C and treated with chronic DSS. CXCL2 mRNA expression was also significantly upregulated in mice fed −I3C with chronic DSS, and this increase was mitigated in mice supplemented with I3C (Appendix A). CCL20 trended towards an increase in mice fed −I3C but was decreased with the addition of I3C (Appendix A). Furthermore, we found that mice induced with chronic colitis and fed −I3C diet had higher MPO activity than mice with DSS-induced with chronic colitis and fed +I3C diet (Appendix A). Further investigation of the immune response in splenocytes of mice fed +I3C and treated with chronic DSS revealed a significant upregulation in mRNA levels of T_reg_ cell marker FOXP3 (Appendix A), as well as a significant increase in the wound-healing gene TGFβ (Appendix A). Taken together, these data demonstrate that the depletion of dietary AhR ligands exacerbates chronic intestinal inflammation, whereas the dietary supplementation of the AhR ligand I3C significantly decreased the severity of inflammation.

The tight junction integrity in the distal colon was next examined (Figure 6E–H). Occludin was significantly decreased in mice fed −I3C and treated with DSS compared to its control (Figure 6E,F). Remarkably, Occludin expression was restored by adding dietary I3C and in DSS-treated mice (Figure 6E,F). Claudin-2 protein levels were increased in mice treated with chronic DSS regardless of diet (Figure 6E,G), and E-cadherin had no change between all groups (Figure 6E,H). Together, these data indicate that activation of AhR through dietary I3C is able to restore tight junction expression in chronic colitis. 

### 3.7. Permeability in Intestinal Epithelial Cells Treated with Cytokines and ICZ

During the digestion process, I3C is converted into indole [3,2-b] carbazole (ICZ) and 3′3-diindolylmethane (DIM) [14]. To gain insights into underlying mechanisms, we next investigated the direct role of ICZ on inflammation in a reductionist model lacking contributions from the gut microbiota. Therefore, we utilized an in vitro model of human intestinal epithelial cells, Caco-2, plating them in transwells and treating them basolaterally with a cytokine cocktail of IL-1β, IFNγ, and TNFα as well as the AhR ligand and breakdown substrate of I3C, ICZ. We confirmed AhR activation by ICZ, showing an increase in the mRNA levels of AhR target gene CYP1A1 (Figure 7A) in cells treated with ICZ, which was blunted with the addition of the cytokine cocktail. Next, we looked at the epithelial secreting cytokine, IL-8, and found a significant increase in cells treated with the cytokine cocktail, which was attenuated with the addition of ICZ (Figure 7B). However, cells treated with the cytokine cocktail alone and both the cytokine cocktail and ICZ had similar significant decreases in TEER (Figure 7C). We examined Occludin at the mRNA level and found no differences in the decrease between cytokine-treated cells with and without ICZ supplementation (Figure 7D). These findings suggest that AhR activation by I3C substrates can mitigate IL-8 levels but is not able to attenuate the increase in permeability after cytokine treatments. Since the intestine is a complex physiological system with an interplay of many factors, including gut microbiota, it was also critical to investigate the effects of dietary supplementation or removal of I3C on gut microbial composition in response to chronic colitis. 

### 3.8. Microbiota Analysis of C57BL/6 Mice Fed −/+I3C Diets Induced with Chronic Colitis

To assess the role of the gut microbiota in response to dietary I3C, cecal contents derived from mice fed −I3C or +I3C diets and treated with or without DSS were subject to 16S rDNA sequencing by ZymoBiomics (Irvine, CA, USA). The total abundance of cecal bacteria did not change between groups (Appendix A); however, the bacterial alpha diversity, as assessed by the Shannon and Chao indices, was significantly increased in mice with chronic colitis supplemented with I3C when compared with DSS mice depleted of dietary AhR ligands (Figure 8A,B). Similarly, Fisher Alpha and total observed species showed significant decreases in diversity in mice fed −I3C and treated with DSS (Appendix A), and phylogenetic diversity showed a drastic decrease in mice fed −I3C and treated with DSS, which was recovered with the addition of I3C (Appendix A). The beta diversity assessed by unweighted Unifrac (Figure 8C) revealed that control mice fed −I3C or +I3C diets (purple and green, respectively) are functionally similar, whereas mice treated with DSS and fed the −I3C or +I3C diets (red and blue, respectively) are each clustered separately, demonstrating functionally distinct microbiota between treatments and diets. Notably, mice fed +I3C diet and treated with DSS (blue) are clustered closer to mice treated without DSS and depleted of the dietary ligand (purple), demonstrating the importance of dietary I3C in maintaining microbial diversity. This illustrates the improvement in microbial diversity and functionality in the presence of dietary AhR ligands in chronic DSS treatment. 

We further investigated these differences at various taxonomic levels. The phyla Actinobacterium, which is increased in DSS-treated mice, was significantly decreased in mice fed supplemented with dietary I3C without DSS treatment (Figure 8D). As Actinobacterium had alterations with only I3C, we identified key bacteria belonging to these ranks that are modified under the presence of I3C as assessed by LefSe analysis (Appendix A). For example, the relative abundance of order Clostridales and families Clostridicae and Lachnospriaceae is significantly decreased in mice treated with DSS but is rescued with the addition of I3C (Figure 8E–G). Our analysis in WT mice fed −I3C or +I3C diets revealed significant increases in *Parvibacter caecicola* (*P. caecicola*) in mice fed −I3C diets under basal conditions (Appendix A), illustrating the importance of I3C in suppressing the emergence of pathobionts. All together, these data demonstrate that supplementation of dietary I3C can decrease the abundance of pathobionts such as *P. caecicola*. 

### 3.9. Supplementation of I3C Improves Paneth Cell Function in Mouse Model of Crohn’s Ileitis

Given the promising effects of dietary I3C in a model of colitis and its ability to restore the microbial dysbiosis, it was of interest to examine how dietary I3C supplementation impacts small intestinal inflammation. A SAMP/YitFC mouse model of ileitis, which developed spontaneous inflammation resembling Crohn’s ileitis with age, with onset of disease starting around 10 weeks and full-blown ileal inflammation around 20 to 40 weeks, was used for the studies [23]. AKR/J mice were used as a control. H&E staining of the ileum revealed clinical manifestations of ileitis, including thickening of the muscle layer, neutrophil infiltration, and epithelial tufting, and these abnormalities were similar in SAMP mice in the presence or absence of I3C (Figure 9A). Villi and crypts were measured to further determine the degree of inflammation, as previously established [24]. There was no change in the villus length between diets (Figure 9B); however, we found a significant increase in the crypt length in SAMP mice compared to AKR mice regardless of diet (Figure 9C). Notably, crypt length was partially recovered in SAMP mice supplemented with I3C as compared to the −I3C diet (Figure 9C), indicating that I3C can attenuate some histological features of the pathogenesis of chronic ileitis. 

Next, we examined mRNA expression of the interleukin (IL) cytokine family in the ileum to analyze the level of inflammation at a molecular level. IL-4 is a T_h_2 cell-derived cytokine that elicits immune responses and is detectable at lesion sites in IBD [25]. We performed RT-PCR of IL-4 in ileal mucosa of SAMP and AKR mice fed −I3C or +I3C diets and found a significant increase in IL-4 expression in SAMP mice fed −I3C compared with AKR mice fed −I3C (Figure 9D). Interestingly, SAMP mice fed I3C exhibited an increase in IL-4 levels compared to control AKR mice fed I3C; however, this increase was not significant (Figure 9D). These data demonstrate that depletion of dietary AhR ligands does not adversely alter the cytokine response, and I3C was able to restore the crypt length. 

The potential effects of dietary I3C supplementation on different epithelial cell populations was examined next. Immunofluorescence staining for the Paneth cell marker LYZ1 showed that on the I3C-depleted diet, SAMP mice exhibited a significantly reduced number of Paneth cells compared to AKR mice (Figure 9E,F). Interestingly, this decrease was significantly attenuated with dietary supplementation of I3C (Figure 9E,F), suggesting that indole-3-carbinole may improve Paneth cell-related dysfunctional abnormalities. Overall, the data demonstrated a beneficial impact of dietary I3C on chronic ileitis by maintaining features of histology and Paneth cell expression.

## 4. Discussion

Many patients with IBD use complementary and alternative therapies, such as dietary alterations, to enhance the effectiveness of conventional therapies or to reduce the side effects caused by conventional medicine [26]. Previous studies have recently shown a protective role of AhR activation in intestinal immune cells in intestinal inflammation [7,10,11]. Yoshimatsu et al. investigated the role of AhR signaling in intestinal CD4^+^ T cells and found that AhR activation induced an accumulation of Helios^+^ T_reg_ cells, as well as altered their localization towards the brush border membrane (BBM) in DSS-induced acute colitis [7]. Metidji et al. interrogated the role of AhR in IECs in acute intestinal inflammation using constitutively deleted AhR in IECs, demonstrating an increased susceptibility to *C. rodentium* infection resulting in decreases in differentiated cells, such as Muc2 and Car4, and goblet cells [9]. Another study investigated the constitutive deletion of IEC AhR in acute DSS-induced colitis and also observed a worsened inflammatory response compared to WT mice [13]. Similar to constitutively deleted IEC AhR in acute inflammation, our current studies utilizing mice with inducible deletion of IEC AhR induced with chronic colitis showed increased severity of inflammation while providing novel insights into the underlying mechanism. For the first time, our studies demonstrate a critical role of AhR in IECs in mediating the severity of chronic intestinal inflammation possibly via ERK-dependent mechanisms. Strikingly, the levels of *p*-ERK were significantly upregulated in AhR^ΔIEC^ mice with chronic DSS. Recent studies have also demonstrated increases in the phosphorylation of ERK in mouse models of colitis [27]; however, rigorous studies are warranted to fully elucidate their interactions. Tamoxifen was used to induce conditional KO of AhR from IECs. Indeed, tamoxifen is a chemotherapeutic drug that can exhibit off-target pro-inflammatory effects alone [28]. However, with respect to intestinal inflammation, one study found that tamoxifen alone did not cause intestinal inflammation [29], and another observed anti-inflammatory effects in acute DSS-induced colitis by an estrogen receptor modulator [30]. Similarly, our studies demonstrated tamoxifen alone exhibited no significant effect on the inflammatory parameters (Appendix A) and showed inconclusive response to DSS (Appendix A). For instance, there was no significant change in the colon length or mRNA expression of TNFα (Appendix A) in the distal colon in DSS-treated mice injected with corn oil or tamoxifen. The MPO levels, however, were significantly decreased by tamoxifen compared with DSS-treated mice injected with corn oil (Appendix A). 

With respect to potential mechanisms, we found that inducible deletion of AhR in IECs does not further impact the increase in serum FITC-dextran in mice treated with chronic DSS. Furthermore, we found no differences between WT and AhR^ΔIEC^ mice with DSS in tight junction proteins Occludin and Claudin-2. Strikingly, the serum levels of LPS were significantly increased in AhR^ΔIEC^ mice with DSS, possibly due to an imbalance of the gut microbiota causing increased production of LPS from gram-negative bacteria. In line with these findings, previous studies utilizing AhR-/- mice found no differences between the increase in permeability in WT and AhR-/- mice with acute DSS treatment [31]. 

As opposed to AhR deficiency, AhR activation has been suggested to be protective in modulating intestinal inflammation [11,12,24,25,26]. Previous studies have shown that AhR activation by potent AhR agonists such as 6-Formylindolo [3,2-*b*] carbazole (FICZ) and β-naphloflavone (βNF) improves intestinal permeability and tight junction integrity in in vitro and mouse models of intestinal inflammation [24,26,32]. One study utilized Caco-2 cells treated with EGTA, a calcium chelator that alters permeability, and found that βNF effectively restored the decrease in TEER [33]. FICZ maintained tight junction protein integrity and transepithelial resistance (TER) in mouse and cell models of acute inflammation [31]. Our studies showed that dietary I3C significantly attenuated inflammation and maintained the expression of tight junction protein Occludin in the distal colon of mice with chronic DSS treatment.

To understand how dietary I3C affects the TJ protein expression and ameliorates inflammation, we utilized a reductionist model of intestinal epithelial cells’ monolayers, Caco-2 cells. Contrary to in vivo studies, we found no differences in TEER and Occludin mRNA in Caco-2 cells treated with cytokines and ICZ, the breakdown product of I3C. ICZ, however, attenuated the increase in IL-8 due to cytokine treatment. To gain further insights into the effects of I3C in the complex in vivo tissue, we assessed the composition of gut microbiota. Our data regarding microbial compositions of WT mice treated with chronic DSS and fed standard Chow diets showed that the alpha and beta diversity were significantly different from the chronic DSS mice fed −I3C or +I3C diets as assessed by Shannon Index and unweighted UniFrac (Appendix A). Together, this indicates that the standard Chow diet and AIN-76A diets result in extremely different microbiota, and furthermore, that AIN-76A supplemented with I3C is more similar to Chow.

Busbee et al. previously demonstrated a recovery in the abundances of *Bacteroides acidifaciens* (*B. acidifaciens)* and *Roseburia* bacteria when mice were i.p. injected with I3C in the TNBS colitis model [15]. In our studies, we identified *Parvibacter caecicola* as a pathobiont that is abundant when mice are treated with chronic DSS and, strikingly, when mice are depleted of I3C at basal level, thus demonstrating the importance of dietary AhR ligands. The species *Parvibacter caecicola* (*P. caecicola*) from the phyla Actinobactera has recently been isolated from TNF^ΔARE^ mice (mouse model elevated TNF levels resulting in ileitis) and has been identified as a pathobiont for intestinal inflammation [34]. However, germ-free studies are needed to elucidate the pathogenicity of this species. Furthermore, we observed that WT mice treated with chronic DSS and supplemented with I3C are clustered closer to the mice given tap water in the beta diversity PCoA plot. Moreover, it is intriguing to see that the mice given tap water and fed −I3C cluster closer to the mice treated with DSS and fed +I3C diet, demonstrating the remarkable effect of removal and supplementation of dietary I3C in modulating the gut microbiota in chronic colitis. Overall, these data illustrate the importance of consuming dietary AhR ligands in ameliorating intestinal inflammation and maintaining gut microbial diversity. As inflammation from DSS treatment is localized to the distal colon, these experiments examined the colon, whereas human IBD can affect the small and/or large intestines. To account for this, we also utilized a well-established mouse model of small intestinal inflammation.

To date, the effect of dietary I3C on the level of inflammation or epithelial abnormalities seen in chronic ileitis has not been investigated. To address this gap in knowledge, we used SAMP/YitFC mice, which develop spontaneous ileitis as they age [23]. In our studies, SAMP1/YitFc mice displayed histological signs of inflammation, including infiltration of inflammatory cells and thickening of the muscle layer. Previous in vitro studies showed that I3C treatment in mouse-derived intestinal organoids promotes the expression of lysozyme, the lineage-specific gene for Paneth cells [35,36]. Lysozyme-positive granules that reside in the cytoplasm of Paneth cells serve as a functional marker for identifying the cell population [35]. In both patients with Crohn’s disease and the TNF^ΔARE^ mouse model of chronic ileitis, severity of inflammation has been correlated with loss of LYZ+ Paneth cells [34,36,37,38]. Animal models of terminal ileitis, like the SAMP1/YitFc mouse, also exhibit Paneth cell alterations [38,39,40,41]. Our immunofluorescent staining for LYZ showed a significant decrease in LYZ+ PCs in the crypts of I3C-depleted SAMP1/YitFc mice, with I3C supplementation resulting in a significant increase in expression. Previous work has reported that these mice also display abnormal Paneth cells and changes in cellular localization, although this was not observed in our studies [40,41]. It is yet to be elucidated whether these Paneth cell abnormalities also lead to the development of CD or increased susceptibility to enteric infections. Together, these data demonstrate that supplementation of I3C in the diet is capable of attenuating pathologies of the ileum, as well as maintaining the expression of Paneth cell marker Lyz1 in ileitis.

Previous studies investigating the impact of dietary AhR ligands used a different base diet, AIN-93G [42], whereas we used the base diet AIN-76A. AIN-76A is purified of all AhR ligands and uses cellulose as the main source of fiber, which is an insoluble fiber, whereas AIN-93G uses dextran, a soluble fiber [43]. Insoluble fibers are directly metabolized into glucose, whereas soluble fibers are first metabolized in short-chain fatty acids (SCFAs) such as butyrate before metabolism into glucose [43,44]. Butyrate in particular is an AhR activator, and it induces the expression of the target gene, IL-22, in immune cells by inhibiting histone deacetylation (HDAC) on the XRE via AhR binding [45,46]. The depletion of not only exogenous dietary ligands but also beneficial endogenous substrates, such as butyrate, could also explain the high mortality rate in AhR^ΔIEC^ mice fed −I3C diet with chronic DSS treatment. Our studies utilize 200 ppm I3C, which is approximately equivalent to 0.1% of I3C in one serving of brussels sprouts, given that the mice consume 5 g of food daily [47]. These data suggest that the consumption of I3C vegetables along with medications can efficiently promote the reduction of inflammation in patients with IBD.

## 5. Conclusions

In conclusion, our findings demonstrate critical roles of IEC AhR in modulating the severity of inflammation and of AhR activation by dietary I3C in maintaining gut microbial composition, suppressing the pathobiont *P. caecicola*, and ameliorating chronic intestinal inflammation. Specifically, we found that deletion of AhR in only IECs exacerbates inflammation, and when dietary ligands are removed, it becomes fatal in mice with chronic DSS-induced colitis. Furthermore, we show that supplementation of the AhR ligand I3C in the diet can attenuate inflammation and restore microbial diversity in chronic colitis. Finally, we illustrate the beneficial impact of consuming dietary I3C on Paneth cell expression in a mouse model of ileitis.

## Figures and Tables

**Figure 1 nutrients-15-04980-f001:**
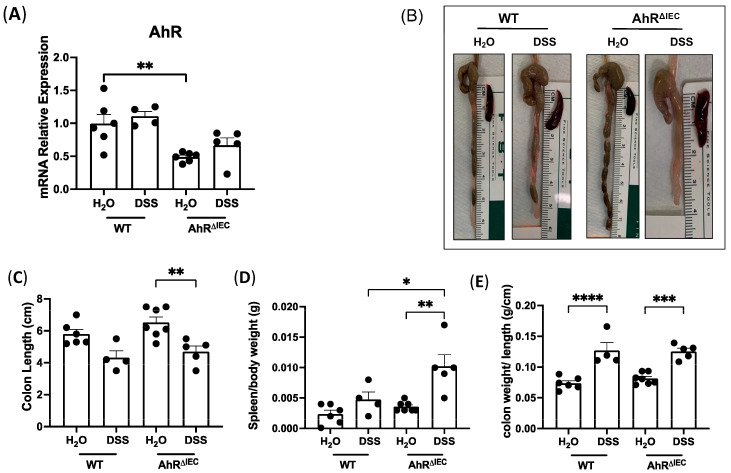
Effect of chronic DSS on mice with inducible deletion of AhR in IECs. AhR^fl/fl^ × Villin-Cre^ERT2^ mice were injected with tamoxifen (AhR^ΔIEC^) for IEC-specific deletion of AhR or the vehicle control (WT) and treated with chronic DSS. AhR deletion was confirmed by (**A**) mRNA levels of AhR normalized to GAPDH in the distal colon. (**B**) Photographs of colons after chronic DSS treatment and quantifications of (**C**) colon length (cm), (**D**) spleen/body weight and (**E**) colon weight/length (g/cm). Values are mean ± standard error of mean. Data were analyzed by one-way ANOVA; * *p* < 0.05, ** *p* < 0.01, *** *p* < 0.001, **** *p* < 0.0001.

**Figure 2 nutrients-15-04980-f002:**
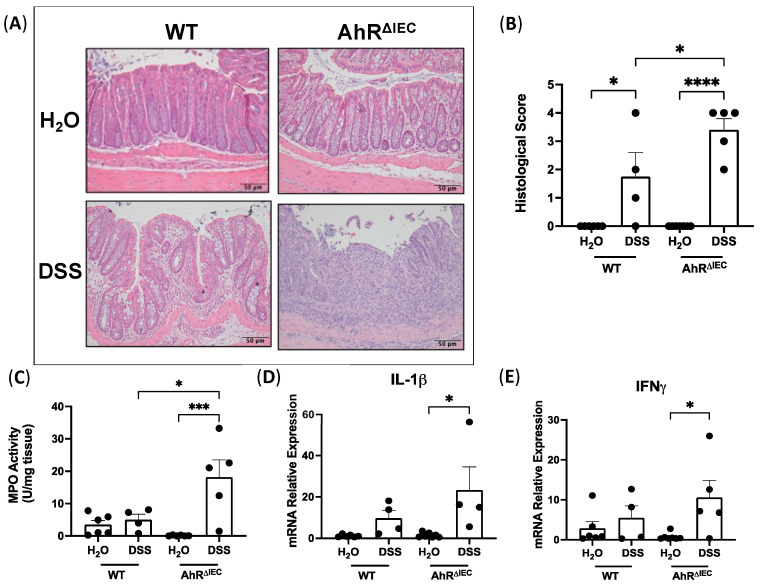
Deletion of AhR in IECs exacerbates DSS-induced chronic colitis in mice. (**A**) H&E stains of distal colon of WT and AhR^ΔIEC^ mice, (**B**) their histological scores, and (**C**) MPO activity. mRNA expression normalized to GAPDH of (**D**) IL-1β and (**E**) IFNγ. Histological scores: 0—no inflammation, 1—quiescent, 2—mildly active, 3—moderately active, 4—severe. Values are mean ± standard error of mean. Data were analyzed by one-way ANOVA; * *p* < 0.05, *** *p* < 0.001, **** *p* < 0.0001.

**Figure 3 nutrients-15-04980-f003:**
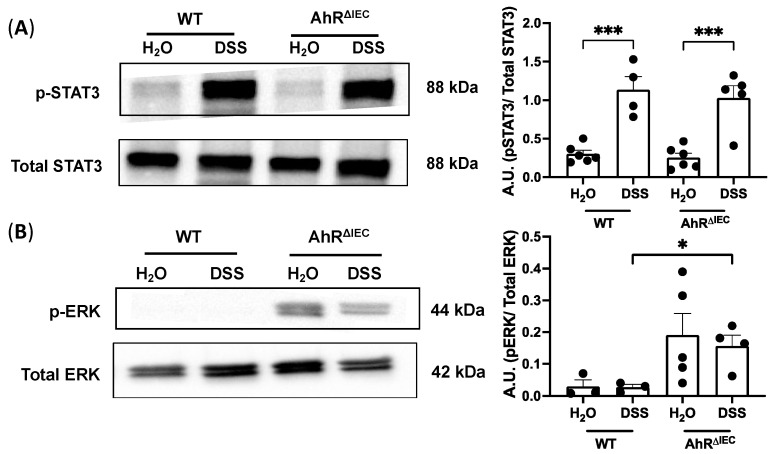
Phosphorylated STAT3 and phosphorylated ERK levels in WT and AhR^ΔIEC^ mice. Western blots of (**A**) phosphorylated STAT3 normalized to total STAT3 and (**B**) phosphorylated ERK normalized to total ERK. Values are mean ± standard error of mean. Data were analyzed by one-way ANOVA; * *p* < 0.05, *** *p* < 0.001. A.U., arbitrary units.

**Figure 4 nutrients-15-04980-f004:**
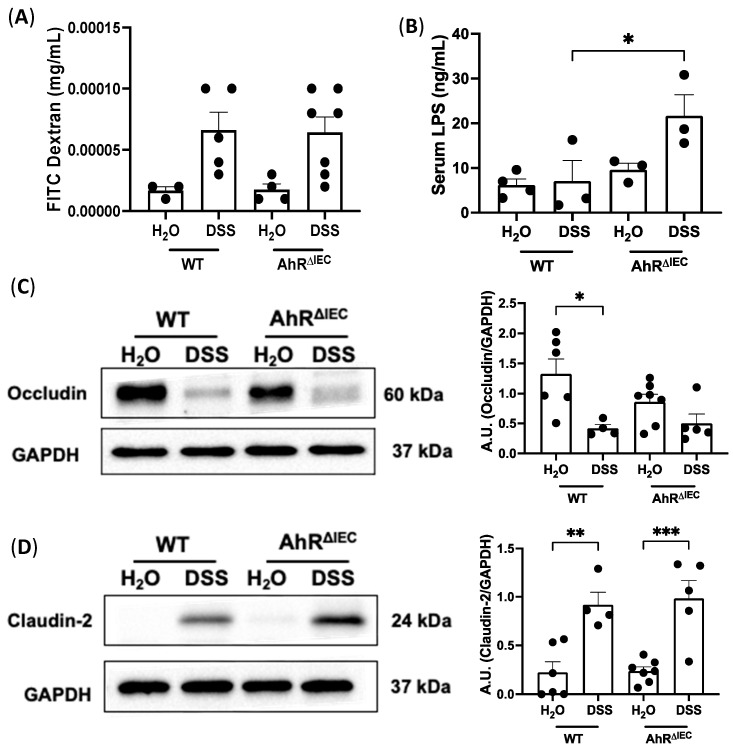
Intestinal permeability in WT and AhR^ΔIEC^ mice treated with chronic DSS. Intestinal permeability in WT and AhR^ΔIEC^ mice induced with chronic colitis was assessed by (**A**) detection in serum of FITC Dextran and (**B**) lipopolysaccharide (LPS) detection in serum. Western blot and quantification of tight junction proteins (**C**) Occludin and (**D**) Claudin-2. Values are mean ± standard error of mean. Data were analyzed by one-way ANOVA; * *p* < 0.05, ** *p* < 0.01, *** *p* < 0.001. A.U., arbitrary units.

**Figure 5 nutrients-15-04980-f005:**
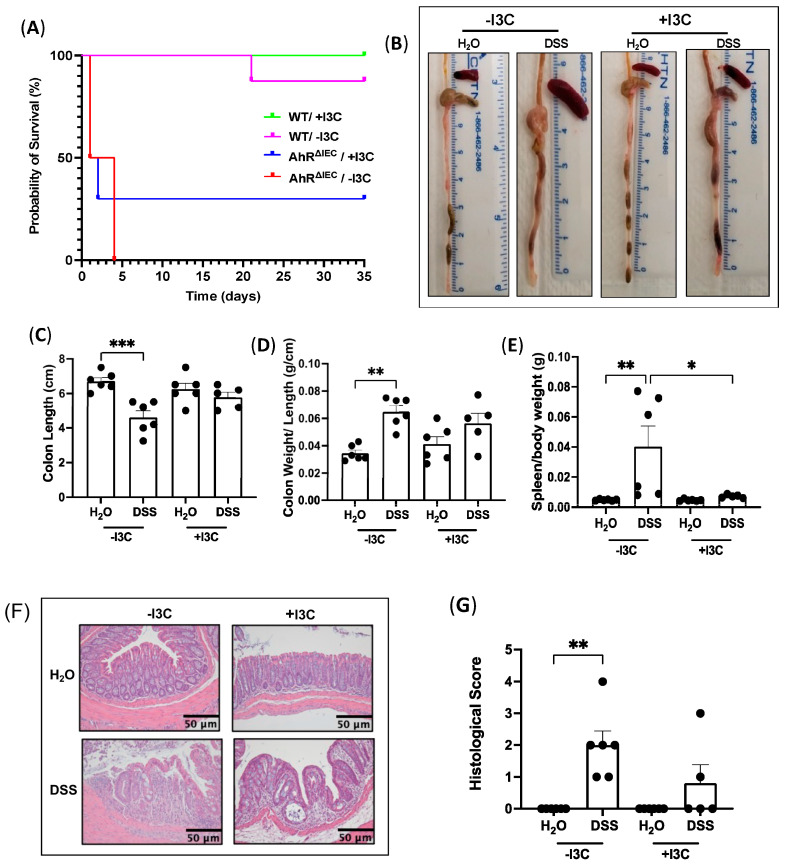
Addition of dietary I3C rescues AhR^ΔIEC^ mice induced with chronic DSS, and removal worsens inflammation phenotypes in WT mice. WT and AhR^ΔIEC^ mice were induced with chronic DSS and fed a diet purified of all AhR ligands (−I3C) or the purified diet supplemented with I3C (200 ppm, +I3C). (**A**) Survival curve of WT and AhR^ΔIEC^ mice induced with chronic DSS and fed −I3C or +I3C diets. WT mice treated with or without DSS and fed −I3C or +I3C diets. (**B**) Photographs of colons of WT mice after consuming −I3C or +I3C diets and DSS treatment and quantifications of (**C**) colon length (cm), (**D**) colon weight/length (g/cm), and (**E**) spleen/body weight (g). (**F**) H&E stains of distal colon and (**G**) their histological scores. Histological scores: 0—no inflammation, 1—quiescent, 2—mildly active, 3—moderately active, 4—severe. Values are mean ± standard error of mean. Data were analyzed by one-way ANOVA; * *p* < 0.05, ** *p* < 0.01, *** *p* < 0.001.

**Figure 6 nutrients-15-04980-f006:**
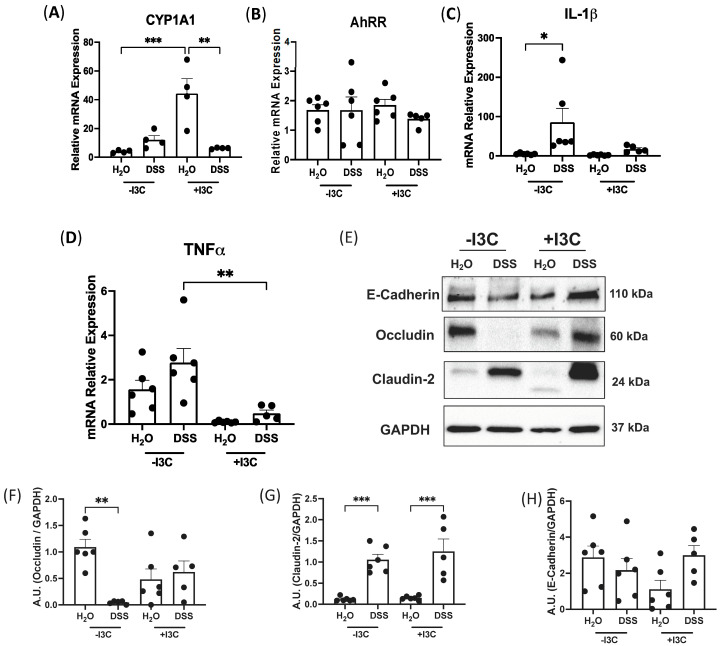
Supplementation of dietary I3C attenuates DSS-induced colitis in WT mice. WT mice were induced with chronic colitis and fed −I3C or +I3C diets. mRNA levels normalized to GAPDH of AhR target genes (**A**) CYP1A1 and (**B**) AhRR and proinflammatory cytokines (**C**) IL-1β and (**D**) TNFα. (**E**) Western blots of tight junction proteins and quantifications of (**F**) Occludin, (**G**) Claudin-2, and (**H**) E-Cadherin, all normalized to GAPDH. Values are mean ± standard error of mean. Data were analyzed by one-way ANOVA; * *p* < 0.05, ** *p* < 0.01, *** *p* < 0.001. A.U., arbitrary units.

**Figure 7 nutrients-15-04980-f007:**
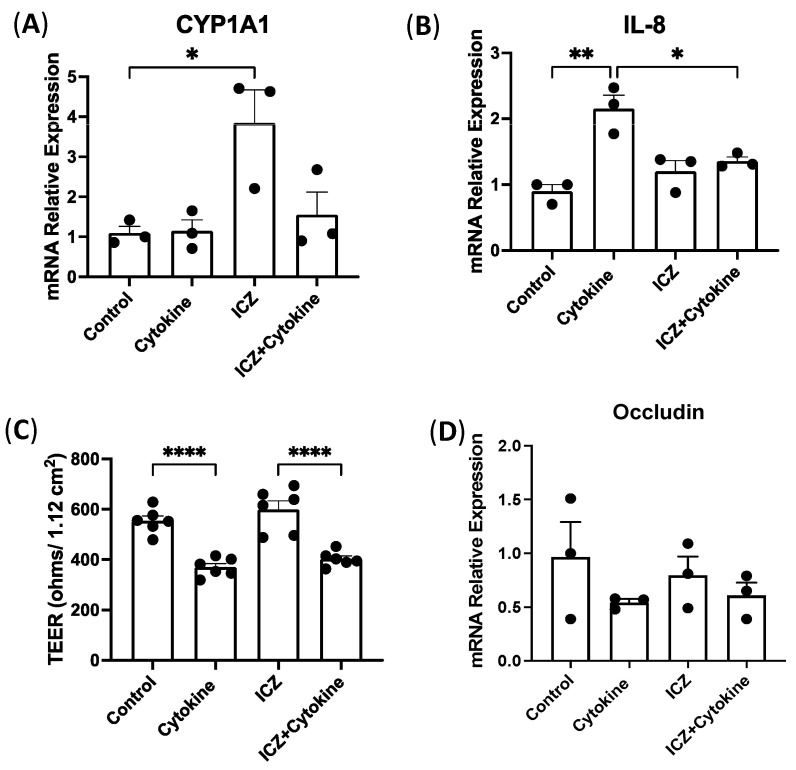
Effect of ICZ on model intestinal epithelial cells treated with cytokines. Caco-2 cells plated in 12-well microporous transwells were treated in the basolateral compartment with a cytokine cocktail of IL-1β (50 ng/mL), IFNγ (30 ng/mL), and TNFα (50 ng/mL). mRNA levels normalized to GAPDH of (**A**) CYP1A1 and (**B**) IL-8. Changes in permeability were measured by (**C**) transepithelial electrical resistance (ohms) and (**D**) mRNA levels of Occludin normalized to GAPDH. Values are mean ± standard error of mean. Data were analyzed by one-way ANOVA; * *p* < 0.05, ** *p* < 0.01, **** *p* < 0.0001.

**Figure 8 nutrients-15-04980-f008:**
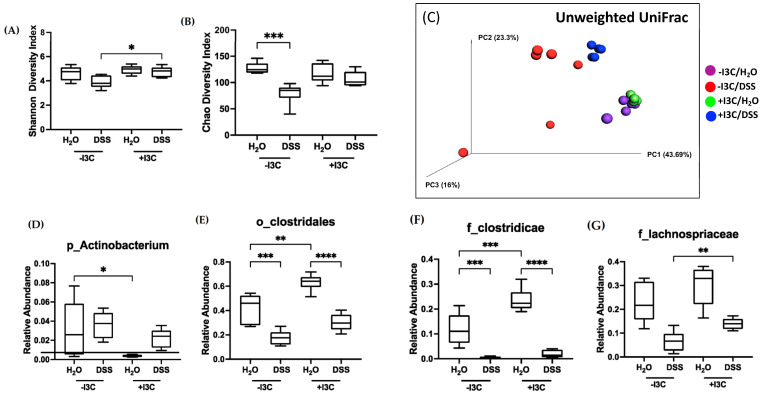
Microbiota analysis of WT mice induced with chronic colitis and fed −I3C or +I3C diets. Cecal contents derived from WT mice induced with chronic colitis and fed the −I3C or +I3C diet were subject to 16S sequencing. Alpha diversity was assessed by (**A**) Shannon index and (**B**) Chao index. Beta diversity was assessed by (**C**) unweighted UniFrac. Relative abundance of (**D**) phylum Actinobacterium, (**E**) order Clostridales, (**F**) family Clostridicae, and (**G**) family Lachnospriaceae. Values are mean ± standard error of mean. Data were analyzed by one-way ANOVA; * *p* < 0.05, ** *p* < 0.01, *** *p* < 0.001, **** *p* < 0.0001.

**Figure 9 nutrients-15-04980-f009:**
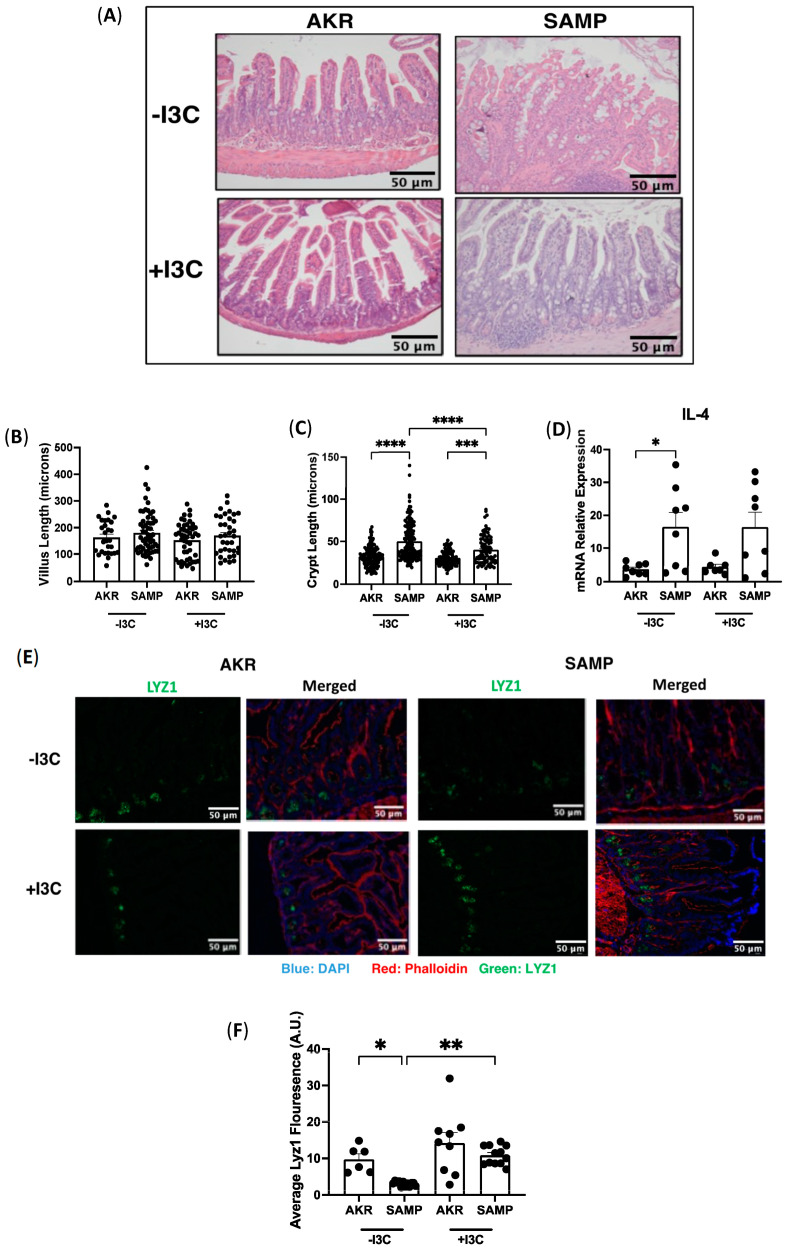
Supplementation of I3C in SAMP/YitFC mice. SAMP/YitFC mice (spontaneous ileitis) and AKR/J (control) were fed −I3C or +I3C diet for 12 weeks. (**A**) H&E staining of ileum from AKR and SAMP mice fed −I3C or +I3C. Histological features were determined by (**B**) villus length and (**C**) crypt length. (**D**) mRNA expression of proinflammatory cytokines IL-4 normalized to GAPDH. (**E**) Immunofluorescence staining of Lyz1 in ileum of AKR and SAMP mice fed −I3C or +I3C diets and (**F**) quantification of fluorescence. Values are mean ± standard error of mean. Data were analyzed by one-way ANOVA; * *p* < 0.05, ** *p* < 0.01, *** *p* < 0.001, **** *p* < 0.0001.

**Table 1 nutrients-15-04980-t001:** Primer sequences used for RT-PCR. All sequences are 5′-3′. Base pairs (bp).

Primer	Host	Forward Sequence 5′→3′	Reverse Sequence 5′→3′	Amplicon Length
AhR	Mouse	GGTACAAGTGCACAATGCCTGC	CAGTGGAATAAGGCAAGAGTGA	44 bp
CYP1A1	Mouse	GGGTTTGACACAGTCACAA	GGGACGAAGGATGAATGCC	38 bp
CYP1A1	Human	TCGGCCACGGAGTTTCTTC	GGTCAGCATGTGCCCAAT	37 bp
CYP1B1	Mouse	CACCAGCCTTAGTGCAGACAG	GAGGACCACGGTTTCCGTGG	41 bp
AhRR	Mouse	GTGGGTTACGATGGACTCAAGG	GTCCCCTGAACAGTGAAATGC	44 bp
IL-4	Mouse	TGATGGGTCTCAGCCCCACCTTGC	CTTTCAGTGTTGTGAGCGTGGACTC	49 bp
IL-8	Human	ATGACTTCCAAGCTGGCCGTGGCT	TCTCAGCCCTCTTCAAAACTTCTC	48 bp
IL-22	Mouse	CGATCTCTGATGGCTGTCCT	ACGCAAGCATTTCTCAGAGA	40 bp
IL-1β	Mouse	GCAACTGTTCCTGAACTCAAC	ATCTTTTGGGGTCCGTCAACT	42 bp
TNFα	Mouse	TACTGAACTTCGGGGTGATTGGTCC	CAGCCTTGTCCCTTGAAGAGAACC	49 bp
IFNγ	Mouse	ATGAACGCTACACACTGCATC	CCATCCTTTTGCCACTTCCTC	42 bp
CXCL2	Mouse	CGCTGTCAATGCCTGAAGAC	ACACTCAAGCTCTGGATGTTCTT	43 bp
CCL20	Mouse	CGACTGTTGCCTCTCGTCACA	GAGGAGGTTCACAGCCCTTT	41 bp
Occludin	Human	ACTTCAGGCAGCCTCGTTAC	GCCAGTTGTGTAGTCTGTCTCA	42 bp
GAPDH	Mouse	TGTGTCCGTCGTGGATCGA	CCTGCTTCACCACCTCTTGAT	40 bp
GAPDH	Human	GAAATCCCATCACCATCTT	AAATGAGCCCCAGCCTTCT	38 bp

## Data Availability

The authors confirm that the data supporting the findings of this study are available within the article and the Appendix A.

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
