# Peer review of "Potential Dietary and Therapeutic Strategies Involving Indole-3-Carbinole in Preclinical Models of Intestinal Inflammation"

_nutrients, 2023, doi:10.3390/nu15234980_

Round 1
Reviewer 1 Report
Comments and Suggestions for Authors
The title doesn't make any sense. I don't understand where the words “gut microbiota” are connected?
L 43-44: “Activation of AhR is crucial for xenobiotic detoxification and the maintenance of intestinal homeostasis”. Reference/s?
L 51: IEC? Please write all the words.
L 63-67: “Given the translational potential of dietary modifications for IBD, we utilized unique approaches involving inducible deletion of AhR from IECs and dietary I3C manipulations models of IBD resembling human IBD. Our data strongly highlight how diet can shape the inflammatory response and microbiota, as well as emphasizes exploiting nutritional strategies to improve chronic intestinal inflammation.” The specific expression belongs to the conclusions and not to the position of the purpose of the research. Please rephrase.
L 99: EMEM. Please write all the words. Same L 119: H&E.
L 128: “Blood was centrifuged at 4,000-5,000 rpm for 10 minutes”. At what temperature?
L 132: OCT?
L 134: “Slides were mounted with Dapi….”. Please explain.
L 138: “Trizol was used for RNA extractions from mouse distal colon….”. Please analyze the methodology obtained RNA extractions.
Table 1. Please complete the table with the following: the name of the primer, Amplicon length (bp), and Reference/s.
L 143: RLT and BME. Please explain.
L 149-157: Reference/s?
L 158-161: Subsection Microbiota. The authors should create a small paragraph, containing the following: Data Analysis and Bioinformatics. Which platform did they use? Were sequences trimmed or excluded as expected errors? About Operational taxonomic units (OTUs), in which similarities were clustered? Did you remove any OTUs with low relative abundance? Samples were screened for chimeras? Accession numbers for sequencing analyses.
L 162-165: Statistical Analysis. The chapter does not cover all necessary statistical analyses of all experimentation.
L 569: “a critical role…..”. Please explain.
L 606: B. acidifaciens. Please write the full name of the bacterium.
L611: “Parvibacter caecicola (p. caecicola)”. Please fix it.
The section Conclusions needs enrichment and same data addition.
Please discuss the methodological limitations of the research and future aspects.
Author Response
We appreciate the reviewer's comments and have made changes to the manuscript accordingly. Below are the reviewer's comments (in bold) and our response (normal type font).
The title doesn't make any sense. I don't understand where the words “gut microbiota” are connected?
Thank you for the comment. We have now removed the words “gut microbiota” from the title.
L 43-44: “Activation of AhR is crucial for xenobiotic detoxification and the maintenance of intestinal homeostasis”. Reference/s?
Reference has been added.
L 51: IEC? Please write all the words.
We corrected it to the proper annotations.
L 63-67: “Given the translational potential of dietary modifications for IBD, we utilized unique approaches involving inducible deletion of AhR from IECs and dietary I3C manipulations models of IBD resembling human IBD. Our data strongly highlight how diet can shape the inflammatory response and microbiota, as well as emphasizes exploiting nutritional strategies to improve chronic intestinal inflammation.” The specific expression belongs to the conclusions and not to the position of the purpose of the research. Please rephrase.
The phrasing has been changing (Line 63-64)
L 99: EMEM. Please write all the words. Same L 119: H&E.
Both abbreviations were corrected.
L 128: “Blood was centrifuged at 4,000-5,000 rpm for 10 minutes”. At what temperature?
Blood was centrifuged for 4000-5000 rpm at 4°C for 10 minutes. This information was added accordingly.
L 132: OCT?
Abbreviation was corrected.
L 134: “Slides were mounted with Dapi….”. Please explain.
Slides were mounted with Dapi to stain for nuclei. This was added in the text (line 142).
L 138: “Trizol was used for RNA extractions from mouse distal colon….”. Please analyze the methodology obtained RNA extractions.
We explained in more detail (Line 144-146)
Table 1. Please complete the table with the following: the name of the primer, Amplicon length (bp), and Reference/s.
Thank you for the suggestion We obtained all our primer sequences from BLAST on the NCBI website and this has been references in the text. We have also added the amplicon length to the table.
L 143: RLT and BME. Please explain.
Details were added in the text (line 152).
L 149-157: Reference/s?
References have been added.
L 158-161: Subsection Microbiota. The authors should create a small paragraph, containing the following: Data Analysis and Bioinformatics. Which platform did they use? Were sequences trimmed or excluded as expected errors? About Operational taxonomic units (OTUs), in which similarities were clustered? Did you remove any OTUs with low relative abundance? Samples were screened for chimeras? Accession numbers for sequencing analyses.
Thank you for the comments. We have added details regarding microbiota bioinformatics in the microbiota section (lines 171-176)
L 162-165: Statistical Analysis. The chapter does not cover all necessary statistical analyses of all experimentation.
We have reviewed the details and changed it accordingly.
L 569: “a critical role…..”. Please explain.
We have modified the text to make it more clear.
L 606: B. acidifaciens. Please write the full name of the bacterium.
We added in the full name.
L611: “Parvibacter caecicola (p. caecicola)”. Please fix it.
Thank you, we have now fixed it.
The section Conclusions needs enrichment and same data addition.
We have updated the conclusion (lines 696-703)
Please discuss the methodological limitations of the research and future aspects.
Thank you for the comment. We have added details in the discussion (line/s 649, 657-659).
Reviewer 2 Report
Comments and Suggestions for Authors
This study shows an important role of AhR activation by dietary I3C in modulating the gut microbiota and chronic intestinal inflammation. Overall, the manuscript addresses an important research question, employs appropriate methodology, and presents valuable insights. However, I have a few minor comments as mentioned below:
1. Are there other studies on the therapeutic potential of AhR and dietary ligands other than I3C? If so, appropriate references should be provided.
2. MPO assay should be explained in more detail. For eg. at what wavelength the absorbance was measured?
3. Line 430: what is the reason for using a cocktail of IL-1β, IFNγ, and TNFα rather than individual cytokines
4. In figure 7, c) and d) are mislabeled.
5. Acronym IEC should be elaborated at first mention.
6. Figure 9e labeling is missing and the scale bars are not visible.
7. While examining upstream factors associated with inflammation (Figure 3), was NF-kB, JNK, p-38 explored as well? If not, provide explanations.
8. Was a non-parametric test such as Kruskal–Wallis test employed for comparing the histological staining in Figure 5F&G?
Author Response
Thank you for the comments and questions. We have addressed the questions below, with the reviewer comments in bold and our response directly below it.
- Are there other studies on the therapeutic potential of AhR and dietary ligands other than I3C? If so, appropriate references should be provided.
b-Naphoflavone (BNF) is a flavonoid which can be found in fruits and vegetables and has been shown to alleviate intestinal inflammation by IP injections in vivo and restore intestinal permeability in vitro. Also, dietary tryptophan has been shown to attenuate DSS-induced colitis in mice. We have made these points clearer in the text (lines 60-62).
- MPO assay should be explained in more detail. For eg. at what wavelength the absorbance was measured?
The details were added in the text (line 130)
- Line 430: what is the reason for using a cocktail of IL-1β, IFNγ, and TNFα rather than individual cytokines
To ensure a strong, general inflammatory response, we utilized a cytokine cocktail.
- In figure 7, c) and d) are mislabeled.
Thank you. This has been fixed.
- Acronym IEC should be elaborated at first mention.
This has been fixed (line 50).
- Figure 9e labeling is missing and the scale bars are not visible.
Thank you. We fixed this and made the scale bars visible.
- While examining upstream factors associated with inflammation (Figure 3), was NF-kB, JNK, p-38 explored as well? If not, provide explanations.
No, these upstream factors were not examined, and we plan to investigate these specific pathways in a separate investigation rigorously utilizing pharmacological inhibitors, biochemical, and cellular imaging approaches.
- Was a non-parametric test such as Kruskal–Wallis test employed for comparing the histological staining in Figure 5F&G?
No, we performed One-Way ANOVA statistical test to compare the histological scores.
Reviewer 3 Report
Comments and Suggestions for Authors
Qazi et al. performed a series of interesting experiments to investigat the functional roles of AHR in intestinal epithelial cells (IECs) in IBD mouse models. The authors found that AhR deletion in IECs exacerbated inflammatory phenotype in chronic colitis. Additionally, they also estimated the protective effect of I3C when supplemented in colonic colitis mice.
Comments:
In general, IECs turnover rate in human gut is 3-5 days, and it is even faster in the mouse GI tract. Several major concerns regarding the experimental design have raised and need to be addressed properly:
1. A precise confirmation of AHR depletion specifically in IECs, preferably through epithelial cells immunohistochemistry rather than a whole mucosa tissue RT-PCR, is warranted.
2. The valid ‘window’ to evaluate AHR depletion should be assessed. The timeline of when the gut tissue was harvested post-tamoxifen injection was not described in the method.
3. Tamoxifen alone can cause epithelium inflammation. Control experiments in WT mice with tamoxifen and corn oil are necessary.
4. Using DSS (1.0-2.5%) for 5 weeks, a chronic induced-colitis model, probably not a valid approach to evaluate AhR function in IECs. An acute DSS-induced would be a better model to address this research question.
Comments on the Quality of English Language
NA
Author Response
Thank you so much for the comments and suggestions. We have updated the manuscript text and supplementary figures. Please see our response below your comments (in bold).
In general, IECs turnover rate in human gut is 3-5 days, and it is even faster in the mouse GI tract. Several major concerns regarding the experimental design have raised and need to be addressed properly:
Thank you for your comments. We agree with the reviewer that IEC turnover in the gut is fast. Thus, we utilized valid approaches for efficient knockdown of AhR from IECs by giving repeated injections of tamoxifen as described previously (Jackson DN, Panopoulos M, Neumann WL, Turner K, Cantarel BL, Thompson-Snipes L, et al. Mitochondrial dysfunction during loss of prohibitin 1 triggers Paneth cell defects and ileitis. Gut 2020;69:1928–38. https://doi.org/10.1136/gutjnl-2019-319523.). The new data presented below addresses the concerns raised.
- A precise confirmation of AHR depletion specifically in IECs, preferably through epithelial cells immunohistochemistry rather than a whole mucosa tissue RT-PCR, is warranted.
We have previously confirmed AhR deletion in both the intestinal mucosa and isolated IECs. Our data demonstrates a significant decrease in both the mucosa and IECs. This has been added in the text and as supplementary figure S1A.
- The valid ‘window’ to evaluate AHR depletion should be assessed. The timeline of when the gut tissue was harvested post-tamoxifen injection was not described in the method.
Thank you for the comment. We re-inject tamoxifen in our mice every two weeks and this is now described in the text.
- Tamoxifen alone can cause epithelium inflammation. Control experiments in WT mice with tamoxifen and corn oil are necessary.
Thank you for your comment. Previous studies have investigated the effect of estrogen and tamoxifen on intestinal inflammation. One study demonstrated that selective estrogen receptor modulators, similar to tamoxifen, alleviated DSS-induced colitis (Polari L, Anttila S, Helenius T, Wiklund A, Linnanen T, Toivola DM, et al. Novel Selective Estrogen Receptor Modulator Ameliorates Murine Colitis. Int J Mol Sci 2019;20:3007. https://doi.org/10.3390/ijms20123007.)
Another study showed that tamoxifen alone did not cause intestinal inflammation as seen through MPO activity and histological scores (Verdú EF, Deng Y, Bercik P, Collins SM. Modulatory effects of estrogen in two murine models of experimental colitis. American Journal of Physiology-Gastrointestinal and Liver Physiology 2002;283:G27–36. https://doi.org/10.1152/ajpgi.00460.2001).
We performed our own studies using C57BL6/J mice injected with corn oil or tamoxifen and induced with DSS-colitis. Similarly, our studies demonstrated tamoxifen alone exhibited no significant effect on the inflammatory parameters (Supple. Figure S4A-B) and showed inconclusive response to DSS (Supple. Figure S4C). There was no significant no change in the colon length and mRNA expression of TNFa (Supple. Figure S4A-B) in the distal colon in DSS-treated mice injected with corn oil or tamoxifen. However, the MPO levels were significantly decreased by tamoxifen compared with DSS-treated mice injected with corn oil (Supple. Figure S4C).
- Using DSS (1.0-2.5%) for 5 weeks, a chronic induced-colitis model, probably not a valid approach to evaluate AhR function in IECs. An acute DSS-induced would be a better model to address this research question.
Acute DSS is typically utilized to induce intestinal injury. Our recently published paper demonstrated that mice induced with chronic DSS display a metabolic profile and gut microbial composition similar to human IBD patients, with active and remission periods (Calzadilla N, Qazi A, Sharma A, Mongan K, Comiskey S, Manne J, et al. Mucosal Metabolomic Signatures in Chronic Colitis: Novel Insights into the Pathophysiology of Inflammatory Bowel Disease. Metabolites 2023;13:873. https://doi.org/10.3390/metabo13070873.). In this regard, this proposed manuscript aimed at elucidating the effects of dietary consumption of I3C in intestinal resembling human IBD, which presents with chronic inflammation including periods of remission (water cycles).